# Measuring the α-particle charge radius with muonic helium-4 ions

Julian J. Krauth[1,2,12 ✉], Karsten Schuhmann[3,4], Marwan Abdou Ahmed[5], Fernando D. Amaro[6], Pedro Amaro[7], François Biraben[8], Tzu-Ling Chen[9], Daniel S. Covita[10], Andreas J. Dax[4], Marc Diepold[1], Luis M. P. Fernandes[6], Beatrice Franke[1,13], Sandrine Galtier[8,14], Andrea L. Gouvea[6], Johannes Götzfried[1], Thomas Graf[5], Theodor W. Hänsch[1,11], Jens Hartmann[11], Malte Hildebrandt[4], Paul Indelicato[8], Lucile Julien[8], Klaus Kirch[3,4], Andreas Knecht[4], Yi-Wei Liu[9], Jorge Machado[7], Cristina M. B. Monteiro[6], Françoise Mulhauser[1], Boris Naar[4], Tobias Nebel[1], François Nez[8], Joaquim M. F. dos Santos[6], José Paulo Santos[7], Csilla I. Szabo[8,15], David Taqqu[3,4], João F. C. A. Veloso[10], Jan Vogelsang[1,16], Andreas Voss[5], Birgit Weichelt[5], Randolf Pohl[1,2 ✉], Aldo Antognini[3,4 ✉] & Franz Kottmann[3,4]

The energy levels of hydrogen-like atomic systems can be calculated with great precision. Starting from their quantum mechanical solution, they have been refined over the years to include the electron spin, the relativistic and quantum field effects, and tiny energy shifts related to the complex structure of the nucleus. These energy shifts caused by the nuclear structure are vastly magnified in hydrogen-like systems formed by a negative muon and a nucleus, so spectroscopy of these muonic ions can be used to investigate the nuclear structure with high precision. Here we present the measurement of two 2S–2P transitions in the muonic helium-4 ion that yields a precise determination of the root-mean-square charge radius of the α particle of 1.67824(83) femtometres. This determination from atomic spectroscopy is in excellent agreement with the value from electron scattering[1], but a factor of 4.8 more precise, providing a benchmark for few-nucleon theories, lattice quantum chromodynamics and electron scattering. This agreement also constrains several beyond-standard-model theories proposed to explain the proton-radius puzzle[2–5], in line with recent determinations of the proton charge radius[6–9], and establishes spectroscopy of light muonic atoms and ions as a precise tool for studies of nuclear properties.

The α particle is the nucleus of the helium-4 ($^4$He) atom and it consists of two protons and two neutrons, tightly bound by the strong nuclear force. It is one of the most-studied atomic nuclei and its properties are of great importance for understanding the nuclear forces and the development of modern nuclear physics[10,11]. Its simplicity makes the α particle a favourable target for a variety of precision studies: its zero nuclear spin ($I = 0$) means that it can be described by a simple charge distribution, without magnetization distribution, quadrupole or higher moments. In elastic electron scattering, the α particle can thus be described by a single charge-monopole form factor. From the analysis of electron scattering world data, a root-mean-square (r.m.s.) charge radius of the α particle $r_\alpha = 1.681(4)$ fm was extracted[1], where the value in brackets indicates the 1$\sigma$ uncertainty in the last digit(s).

In atomic spectroscopy, $I = 0$ results in the absence of any hyperfine structure, which simplifies the interpretation of atomic spectra substantially. However, so far, no determination of $r_\alpha$ exists from atomic spectroscopy. In fact, so far, the only absolute radii determined by laser spectroscopy are for the proton and the deuteron[7,8,12], because the required combination of sufficiently precise measurements and theory calculations exists only for atomic hydrogen (H) and deuterium (D) (atomic number $Z = 1$). For elements with $Z > 1$, laser spectroscopy has yielded only differences of charge radii within an isotopic chain[13–18] by measuring the same atomic transition in various isotopes to eliminate the common energy shifts related to the interaction among electrons. For the determination of absolute radii from He atoms (three-body system with two electrons), theory is not yet advanced enough[19]. Sufficiently precise experiments with the H-like He$^+$ ion, where the two-body

[1]Max Planck Institute of Quantum Optics, Garching, Germany. [2]QUANTUM, Institut für Physik & Exzellenzcluster PRISMA, Johannes Gutenberg-Universität Mainz, Mainz, Germany. [3]Institute for Particle Physics and Astrophysics, ETH Zurich, Zurich, Switzerland. [4]Paul Scherrer Institute, Villigen, Switzerland. [5]Institut für Strahlwerkzeuge, Universität Stuttgart, Stuttgart, Germany. [6]LIBPhys-UC, Department of Physics, University of Coimbra, Coimbra, Portugal. [7]Laboratory for Instrumentation, Biomedical Engineering and Radiation Physics (LIBPhys-UNL), Department of Physics, NOVA School of Science and Technology, NOVA University Lisbon, Caparica, Portugal. [8]Laboratoire Kastler Brossel, Sorbonne Université, CNRS, ENS-PSL Research University, Collège de France, Paris, France. [9]Physics Department, National Tsing Hua University, Hsincho, Taiwan. [10]i3N, Universidade de Aveiro, Aveiro, Portugal. [11]Ludwig-Maximilians-Universität, Fakultät für Physik, Munich, Germany. [12]Present address: LaserLaB, Department of Physics and Astronomy, Vrije Universiteit, Amsterdam, The Netherlands. [13]Present address: TRIUMF, Vancouver, British Columbia, Canada. [14]Present address: Institut Lumière Matière, University of Lyon, Université Claude Bernard Lyon 1, CNRS, Villeurbanne, France. [15]Present address: Theiss Research, La Jolla, CA, USA. [16]Present address: Department of Physics, Lund University, Lund, Sweden. ✉e-mail: j.krauth@vu.nl; pohl@uni-mainz.de; aldo@phys.ethz.ch

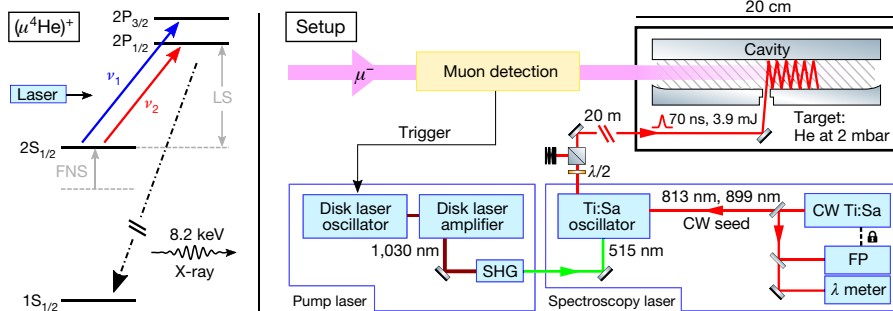

**Fig. 1 | Energy-level scheme and experimental setup.** Left: energy levels of interest in $(\mu^4\mathrm{He})^+$. We drive the 2S → 2P transitions $\nu_1$ and $\nu_2$ (at wavelengths of 813 nm and 899 nm, respectively) and measure the 8.2-keV Lyman-α X-ray from the subsequent decay to the $1S_{1/2}$ ground state. Indicated are the Lamb shift (LS) and the shift due to the finite nuclear size (FNS), which is proportional to $r_\alpha^2$. Right: sketch of the experimental setup (not to scale). On the way to the He target, the muon is detected, thereby triggering the laser system. After the muon is stopped in 2 mbar of He gas at room temperature, $(\mu^4\mathrm{He})^+$ is formed.

About 1 μs after the trigger, the laser pulse arrives at the target, is coupled into the multipass cavity and distributed over the entire muon stop volume (hatched area). The pulse is produced by a Ti:Sa oscillator seeded by a continuous-wave (CW) Ti:Sa laser and pumped by a frequency-doubled pulsed thin-disk laser. The continuous-wave Ti:Sa laser is stabilized to a Fabry–Pérot (FP) cavity and referenced to a wavemeter. The Lyman-α X-rays are measured via LAAPDs (not shown) mounted above and below the cavity. SHG, second harmonic generation.

theory of H is applicable, will soon be available[20,21]. Thus, the present work, with a muonic ion, provides the first, to our knowledge, determination of a nuclear charge radius from laser spectroscopy beyond the proton and the deuteron.

In light muonic atoms and ions, a single muon orbits a bare nucleus. Owing to the large muon mass $m_\mu \approx 200 m_e$, where $m_e$ denotes the electron mass, the muon's Bohr radius is smaller than the electron's Bohr orbit in the corresponding H-like ion by a factor of about 200. This results in a roughly $200^3 \approx 8$ million times increased overlap of the muon's wave function with the nucleus and a correspondingly increased sensitivity to nuclear properties, such as the nuclear charge radius. This finite extension of the nucleus modifies the so-called Lamb shift[22], which is the energy difference between the $2S_{1/2}$ and $2P_{1/2}$ states. Here we present the first measurement of the 2P–2S energy splittings in the H-like muonic He ion $(\mu^4\mathrm{He})^+$. Combined with the corresponding theoretical prediction, our measurement yields a precise determination of the α-particle charge radius $r_\alpha$.

The lowest atomic levels in $(\mu^4\mathrm{He})^+$ are sketched in Fig. 1 (left). The Lamb shift is dominated by pure quantum electrodynamics (QED) effects, in particular, vacuum polarization, which is vastly enhanced in muonic atoms[23] (see Methods), but the effect of the finite nuclear size amounts to as much as 20% of the total energy splitting. Therefore, already a moderately precise measurement of the 2P–2S energy difference can yield a vastly improved value of the α particle's charge radius.

The theoretical expression for the $2P_{1/2}$–2S energy difference in $(\mu^4\mathrm{He})^+$ is given as (see Methods)

$$\begin{aligned}
\Delta E_{2P_{1/2}-2S}^{\mathrm{theo}} = {}& 1{,}668.489(14)\ \mathrm{meV} \\
& - 106.220(8)\ \mathrm{meV\,fm}^{-2} \times r_\alpha^2 + 0.0112\ \mathrm{meV} \\
& + 9.340(250)\ \mathrm{meV} \\
& - 0.150(150)\ \mathrm{meV}.
\end{aligned} \quad (1)$$

The first term is the sum of pure bound-state QED contributions (such as radiative, recoil and relativistic), which are independent of the nuclear structure. The second term is the finite-size effect. It is proportional to the square of the α-particle r.m.s. charge radius $r_\alpha$ and includes mixed radiative–finite-size contributions. The next, small, term in equation (1) is implicitly radius dependent but cannot be parameterized as being proportional to $r_\alpha^2$. As this term is small, it is sufficient to calculate it using electron-scattering results[24]. The fourth term is the two-photon exchange (2PE) term resulting from the sum of the third Zemach moment contribution extracted from electron–proton scattering data[25] and the polarizability

contribution computed using a state-of-the-art ab initio few-nucleon approach[26].

The last term in equation (1) is the sum of the calculated elastic and the unknown inelastic three-photon exchange (3PE) contributions. The former was used to estimate the latter, considering the cancellation of the two terms observed in muonic deuterium[27], as detailed in the Methods. We assign to the sum a conservative 100% uncertainty (1σ).

The 2P fine structure[23]

$$\Delta E_{2P_{3/2}-2P_{1/2}}^{\mathrm{theo}} = 146.1828(3)\ \mathrm{meV} \quad (2)$$

is about half as large as the finite-size effect and can be calculated with great precision due to the absence of both the hyperfine structure and the leading-order nuclear finite-size effects.

The experimental determination of the Lamb shift and the fine structure of $(\mu^4\mathrm{He})^+$ reported here follows the technique of our previous muonic H and muonic D measurements[2,3,6]. About 500 muons per second from the world's most intense beam of negative muons at ultralow energy (a few kiloelectronvolts) at the πE5 beam line of the Paul Scherrer Institute (Switzerland) are stopped in 2 mbar of He gas at room temperature. The slowing down in the He gas occurs through collisions. In the last collision, the muon ejects an electron and gets captured by a He atom, forming a muonic atom in a highly excited state (with principal quantum number $n \approx 14$). For these high-$n$ orbits, the (internal) Auger rates are much larger than the radiative transition rates, and hence the remaining electron is quickly ejected. The resulting H-like ion proceeds within about 100 ns (refs. [28,29]) to the ground 1S or to the metastable 2S state through radiative transitions. Roughly 1% of the muons will populate the metastable 2S state[30] of $(\mu^4\mathrm{He})^+$, whose lifetime of 1.75 μs is a result of muon decay and two-photon 2S → 1S de-excitation. In fact, at our low target gas pressure of only 2 mbar, the 2S → 1S collisional quenching rate is less than 10 kHz (ref. [30]; less than 0.01 quenching probability per microsecond) and with our sufficiently clean target gas, the $(\mu^4\mathrm{He})^+$ ion will not be neutralized.

A pulsed laser system (Fig. 1, right) is triggered on the arrival of a single muon and illuminates the muon stop volume about 1 μs after the muon stop. The laser system comprises a titanium:sapphire (Ti:Sa) oscillator, which is pumped by a frequency-doubled thin-disk laser and injection seeded by a continuous-wave Ti:Sa laser. It is widely tunable from 800 nm to 1,000 nm and delivers pulses of energy up to 10 mJ with a bandwidth of less than 100 MHz. The measurements are, however, performed at a constant pulse energy of 3.9 mJ to avoid power broadening (about 10 mJ and 20 mJ are needed to saturate the 2S → $2P_{3/2}$ and 2S → $2P_{1/2}$ transitions, respectively) and to avoid laser-induced damage

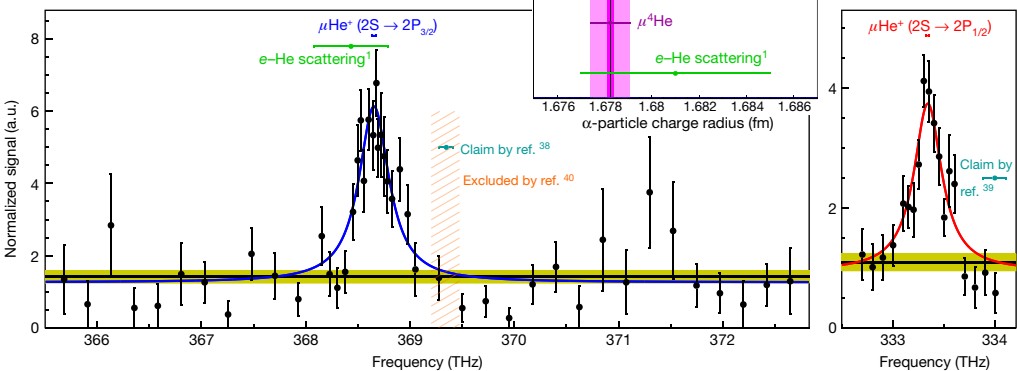

**Fig. 2 | The measured transitions.** The 2S → P$_{3/2}$ (left) and 2S → 2P$_{1/2}$ (right) resonances in ($\mu^4$He)$^+$ fitted with a power-broadened Lorentzian line-shape at a fixed linewidth of 319 GHz (FWHM) given by the 2P lifetime. The black data points show the laser-induced events (number of X-rays in time coincidence with the laser light), normalized to the prompt events (number of X-rays from the cascade on formation of ($\mu^4$He)$^+$). The horizontal band shows the background levels (with 1$\sigma$ uncertainty) obtained from measured data where the laser was not triggered. The tiny bars above the resonances show the 1$\sigma$ uncertainty of the fitted resonance position. Indicated are the erroneous claims from refs. [38,39] the hatched exclusion region from ref. [40] and the expected resonance position using $r_\alpha$ from e–He scattering[1]. The latter is compared to our radius from ($\mu^4$He)$^+$ in the inset where the inner and outer error bands represent the experimental and total uncertainty, respectively (see equation (7)). a.u., arbitrary units.

of the multipass cavity used to enhance the laser fluence in the muon stop volume. Every few hours, the laser frequency is altered by changing the frequency of the continuous-wave seed laser. The energy delivered to the multipass cavity is adjusted with a half-wave-plate ($\lambda/2$) and a polarizer.

The on-resonance laser pulses excite the muonic He ions from the 2S to the 2P state. The Lyman-α X-ray at 8.2 keV emitted by the fast decay of the 2P state into the 1S ground state is detected with large-area avalanche photodiodes (LAAPDs). For the data analysis, we select laser-induced events in which a single muon enters the apparatus, and a muonic X-ray is observed in coincidence with the laser pulse in the cavity. In addition, we require the detection of an electron from muon decay shortly afterwards in either an LAAPD or in a set of plastic scintillators placed around the target, which suppresses the background by about an additional order of magnitude, while maintaining half of the good events. For the LAAPDs, we apply a waveform analysis, which improves energy and time resolution and allows for discrimination between X-rays and megaelectronvolt electrons from $\mu^-$ decay[31].

The two resonances shown in Fig. 2 are obtained by plotting the number of such laser-induced events as a function of laser frequency, normalized by the number of prompt X-rays from the muonic ion formation to account for fluctuations in the muon beam intensity. The 2S → 2P$_{3/2}$ transition was measured first due to its larger matrix element ($M = 1.7 \times 10^{-16}$ cm$^2$) and correspondingly expected larger signal. The data were taken over ten days, which included searching for the resonance over a frequency range of 7 THz. The 2S → 2P$_{1/2}$ transition with $M = 0.8 \times 10^{-16}$ cm$^2$ was then found immediately due to the well known fine structure (equation (2)), and we spent 13 days measuring it.

The two resonances are fitted with a line-shape model, taking into account the saturation fluence and the measured laser pulse energy, which varied slightly over the data-taking period. We find that a fit of a simple Lorentzian produces line centres in agreement with the ones from the line-shape model.

The fitted line widths agree well with the 319 GHz (ref. [32]) expected from the 2P state lifetime, hence the width in the final analysis is fixed. The experimental values of the frequencies we obtain for the two transitions are

$$\nu^{\exp}(2S \to 2P_{3/2}) = 368,653 \pm 18 \text{ GHz} \tag{3}$$

$$\nu^{\exp}(2S \to 2P_{1/2}) = 333,339 \pm 15 \text{ GHz}. \tag{4}$$

The uncertainties mostly stem from statistics (298 and 284 events above background for the two resonances, respectively). This is because our experiment does not suffer from any relevant systematics: usual systematic effects of atomic physics experiments, such as Doppler, Stark and even the Zeeman shifts in our 5-T field (Methods), amount to less than 0.1 GHz. Uncertainties from the laser frequency calibration, including chirp, are of the order of 0.1 GHz. The only conceivable relevant systematic shift would originate from a systematic pulse energy imbalance between measurements on the left wing and right wing of the resonance. As our fit function accounts for variations of the pulse energy, the fitted position is essentially free from this systematic shift. We assign a conservative systematic error of 3 GHz to this effect to account for uncertainties in the pulse energy measurements (Methods).

The difference between the two frequencies in equations (3) and (4) yields the experimentally determined 2P fine structure of

$$\Delta E^{\exp}_{2P_{3/2}-2P_{1/2}} = 146.047 \pm 0.096 \text{ meV}, \tag{5}$$

converted to millielectronvolts using 1 meV = 241.799 GHz. It is less precise but in good 1.4$\sigma$ agreement with the theory value of equation (2). Hence, we use equation (2) to combine our two measured transition frequencies (equations (3) and (4)) and obtain a value for the Lamb shift of

$$\Delta E^{\exp}_{2P_{1/2}-2S} = 1{,}378.521 \pm 0.048 \text{ meV}, \tag{6}$$

which in conjunction with equation (1) gives

$$r_\alpha = 1.67824(13)_{\exp} (82)_{\text{theo}} \text{ fm}. \tag{7}$$

Here the experimental uncertainty of 0.13 attometres (am) is given by statistics. The theory uncertainty by far limits the extraction of $r_\alpha$. Its 0.82-am uncertainty is from 2PE (0.70 am), 3PE (0.42 am, given by our conservative estimate of the inelastic part), QED (0.04 am) and the $r_\alpha^2$ coefficient (0.06 am) in equation (1). The dominant uncertainty is thus from the nuclear and nucleon polarizability contributions to the 2PE of equation (1)[26,33–37]. This uncertainty accounts for variations and truncation of the nuclear potential, numerical convergence, few-body methods, and for the various methods of including the nucleon finite size and relativistic effects.

We note that almost 50 years ago, a group at CERN claimed[38,39] to have measured the transitions presented here, but, as we show, those measurements are wrong. The experiment was conducted at a

20,000-times-higher He gas pressure of 40 bar. Doubts were raised[30] owing to the high collisional quench rate of roughly $6 \times 10^{10}$ s$^{-1}$, equivalent to a 2S state lifetime of only 17 ps. No $(\mu^4\mathrm{He})^+$ ion in the 2S state could possibly have survived the 0.5-μs time delay until the laser pulse arrived. A direct measurement[40] at the Paul Scherrer Institute (PSI) excluded the earlier result by $3.5\sigma$. Nevertheless, the value is still used in the literature. Our measurement shows that there is no signal observed at the position claimed in the earlier experiment ($\nu = 369,350(140)$ GHz) whose resonance is located $5\sigma$ (or 2.2 linewidths) to the right of our peak. We thus confirm that the earlier experiments have been wrong. Intriguingly, their quoted charge radius[38] is not very far from our value, but this can be traced back to an awkward coincidence of a wrong experiment combined with an incomplete 2P–2S theory prediction, by chance yielding a not-so-wrong value of $r_\alpha$.

The $r_\alpha$ value from our measurement is in excellent $0.7\sigma$ agreement with the world average from elastic electron scattering[1] $r_\alpha(\mathrm{scatt}) = 1.681(4)$ fm, but 4.8 times more precise. Our precise charge radius for $(\mu^4\mathrm{He})^+$ can hence be used to constrain and improve fits of the electric form factor. This is in contrast to our previous measurements in muonic H (refs. [2,3]) and muonic D (ref. [6]), which revealed a large and unexpected discrepancy with the proton radius from both electron scattering and H spectroscopy, coined the 'proton-radius puzzle'[4,5].

The agreement between our measurement and the electron-scattering result constrains muon-specific beyond-standard-model effects to contribute less than 3.4 meV (95% confidence level) to the 2P–2S energy splitting in $(\mu^4\mathrm{He})^+$. This upper bound is limited by the uncertainty of the α-particle charge radius $r_\alpha(\mathrm{scatt})$ extracted from electron-scattering experiments. Hence the $(\mu^4\mathrm{He})^+$ measurements exclude the scenarios of ref. [41] to explain the proton-radius puzzle, which predicted a 6.4-meV difference for the $(\mu^4\mathrm{He})^+$ measurement. For the model of ref. [42], we can set an upper limit of 4.5 MeV on the mediator mass. Following equation 23 in ref. [43], our $(\mu^4\mathrm{He})^+$ measurement constrains any beyond-standard-model short-range muon–proton interaction to be smaller than 3.4 meV/20 ≈ 0.17 meV in muonic H. These short-range interactions also include exotic gravitational[44] and hadronic effects related to the subtraction term in the 2PE in muonic hydrogen[45]. However, recognizing the recent measurements in H spectroscopy and electron–proton scattering that yield the smaller 'muonic' proton radius[7–9], there is in fact no need to invoke beyond-standard-model scenarios, but note that a large proton radius was recently obtained in ref. [12]. Still, the $(\mu^4\mathrm{He})^+$ measurement provides interesting bounds for possible flavour-violating short-range interactions between the muon and a proton or a neutron[43,46–48].

The obtained charge radius represents a benchmark for few-nucleon theories[35]. A recent calculation of the form factor obtained from potentials based on chiral effective field theory ($\chi$EFT) yields a charge radius of $r_\alpha = 1.663(11)$ fm (ref. [49]), in good agreement with our value from $(\mu^4\mathrm{He})^+$. Our $r_\alpha$ can also be used to fix low-energy constants of the nuclear potential, that is, to fix, together with nucleon–nucleon scattering data and nuclear binding energies, the two-nucleon and three-nucleon forces derived in a $\chi$EFT framework[50,51].

A test of higher-order bound-state QED is expected by combining our $(\mu^4\mathrm{He})^+$ measurements with upcoming measurements of the 1S–2S transition in ordinary He$^+$ ions[20,21]. In fact, these challenging higher-order contributions scaling with the nuclear charge as $Z^5$ to $Z^7$ are massively increased in He$^+$ compared with H (refs. [20,52–54]). Our charge radius determination allows for a test of these contributions to the 1S Lamb shift in He$^+$ with an accuracy of 60 kHz, corresponding to $6 \times 10^{-12}$ of the 1S–2S transition in He$^+$.

Alternatively, the combination of our measurement with future results for the 1S–2S transition in He$^+$ (refs. [20,21]) will yield a value of the Rydberg constant with 24-kHz accuracy. The Rydberg constant is at present known with a precision of 6 kHz (using CODATA 2018; see https://physics.nist.gov/cuu/Constants), but has recently jumped by about 100 kHz, owing to our measurements in muonic H and muonic

D. A measurement from an atomic system with $Z = 2$ represents a determination independent of experiments in H and muonic H.

With slightly improved uncertainty in the QED calculations for the 1S–2S transition in He$^+$ (currently at about 40 kHz; refs. [52,53]), He$^+$ spectroscopy will yield the α-particle charge radius with an uncertainty a factor of two smaller than our value from $(\mu^4\mathrm{He})^+$. Inserting this $r_\alpha$ value into the Lamb shift prediction for $(\mu^4\mathrm{He})^+$ and comparing it with our experimental result, will yield a value of the summed 2PE and 3PE contributions on the 0.1-meV level. This constitutes a precise benchmark for few-nucleon theories, as at present the 2PE contribution calculated using a phenomenological potential (AV18+UIX) differs by about 0.5 meV from the prediction based on a $\chi$EFT approach[26].

Ultra-precise measurements exist in atomic helium[13,14,16–18], but theory[19] for this two-electron system is not yet advanced enough to allow for an absolute determination of the α-particle size from these measurements. Despite this, there are isotope shift measurements in He atoms that can be used to deduce the charge radius difference between any He isotope $^4$He and the α particle: $r^2(^4\mathrm{He}) - r_\alpha^2$ ($A$, mass number). For the case of $^3$He, several measurements of the isotope shift in regular He atoms exist, but they differ by as much as $9\sigma$ (refs. [13,17]), which renders any $^3$He charge radius determination unreliable. This discrepancy will hopefully be resolved by ongoing measurements[16,17] in $^3$He and our upcoming results on $(\mu^3\mathrm{He})^+$.

Using the new $r_\alpha$ from $(\mu^4\mathrm{He})^+$ as an anchor point for the isotope shift measurements[15] in $^6$He and $^8$He given by $r^2(^6\mathrm{He}) - r_\alpha^2 = 1.415(31)$ fm$^2$ and $r^2(^8\mathrm{He}) - r_\alpha^2 = 1.009(62)$ fm$^2$, we obtain the r.m.s. charge radii of the unstable nuclei $^6$He and $^8$He

$$r(^6\mathrm{He}) = 2.0571(7)_{r_\alpha}(75)_{\mathrm{iso}}\ \mathrm{fm}, \tag{8}$$

$$r(^8\mathrm{He}) = 1.9559(7)_{r_\alpha}(158)_{\mathrm{iso}}\ \mathrm{fm}, \tag{9}$$

respectively, where the first uncertainties are from our new $r_\alpha$ value and the second uncertainties from the electronic isotope shift measurements. Our $r_\alpha$ value hence paves the way for a tenfold improvement in the important $^6$He and $^8$He halo-nuclei.

In conclusion, the precise α-particle charge radius from laser spectroscopy of $(\mu^4\mathrm{He})^+$ serves as a benchmark for few-nucleon theories and for lattice quantum chromodynamics calculations, and can be used to improve the fits of the electric form factor at low-$Q^2$ ($Q$, momentum transfer) of the $^4$He nucleus. Moreover, it serves as an anchor point for isotopic shift measurements, it opens the way to test higher-order bound-state QED contributions to an unprecedented sensitivity when combined with measurements in He$^+$, it can be used for a determination of the Rydberg constant independent of hydrogen and the proton radius[3], and it provides bounds for flavour-violating interactions. Advances of few-nucleon theories and nuclear potentials—including a fully consistent calculation of the 2PE contribution where the α-particle charge radius is used to fix the nuclear potential—can further improve the α-particle charge radius and our understanding of the nuclear structure, with the ultimate potential of disclosing subtle effects such as the nucleon confinement (swelling) in nuclei[55].

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

## Methods

### Creation of slow muons and muonic atom formation

The experiment was conducted at the πE5 area of the high-intensity proton accelerator (HIPA) at the PSI in Switzerland, giving access to a 102 MeV $c^{-1}$ negative pion beam. The pions tangentially enter a cyclotron trap made of two superconducting 4-T coils at a rate of $10^8 s^{-1}$. Here they are moderated, trapped and decay into negative muons and muon antineutrinos. Muons from backwards decaying pions with energies of a few megaelectronvolts are confined in the cyclotron trap, which acts as a magnetic bottle. They are further slowed down by repeatedly passing a 160-nm-thick Formvar foil coated with nickel and installed at the centre of the cyclotron trap. Applying a voltage of −20 kV on the foil, the muons are extracted axially through one of the exits of the magnetic bottle. The muons are then transported to the measurement section (in a region of lower background) via a momentum filtering toroidal magnetic field of about 0.15 T, created by 17 identical magnetic coils and a collimator. The He gas target is located within a 1-m-long solenoid at a 5-T magnetic field, reducing the muon beam radius to a few millimetres. Before the muons enter the target with a rate of about 500 $s^{-1}$, they cross thin carbon foils where they are detected to trigger the laser system. The detection[2] uses electron emission induced by the muons crossing the foils, which also slows them down to few kiloelectronvolts. They are then stopped in the 20-cm-long He gas target (room temperature, pressure of 2 mbar) via ionizing collisions with the He atoms. In its final collision, the muon remains bound to a He nucleus. Starting from a highly excited state ($n \approx 14$), it cascades down ejecting the second electron of the He atom via an Auger process and emitting prompt X-rays. A small fraction of roughly 1% of the muons ends up in the metastable 2S state (lifetime $\tau = 1.75\,\mu s$), which is the initial state needed for our measurement.

### Laser spectroscopy and data analysis

The laser system is optimized for fast pulse delivery upon a trigger signal caused by the stochastic muon arrival. The short latency time of 1 μs is obtained using an Yb:YAG (ytterbium-doped yttrium aluminium garnet) thin-disk laser where the energy is continuously stored in the active medium through continuous-wave pumping with commercial diode lasers of kilowatt optical power. The thin-disk laser consists of a Q-switched oscillator operating in pre-lasing mode followed by an eight-pass amplifier delivering (during data taking) pulses of 80-mJ energy and 20-ns length at 1,030 nm. These pulses are frequency doubled in a lithium triborate crystal to pump a ring Ti:Sa oscillator injection seeded by a tunable (from 800 nm to 1,000 nm) continuous-wave and single-frequency Ti:Sa laser. The spectroscopy pulses, with energies of about 10 mJ and a bandwidth less than 100 MHz, pass through a $\lambda/2$-waveplate and a polarizer allowing adjustment of the pulse energy. The pulses are then transported over 20 m inside an evacuated tube from the laser hut to the muon beamline and eventually coupled into the multipass cavity[56] through a 0.6-mm-diameter hole. The injection into the cavity is optimized using photodiodes monitoring the cavity lifetime and the light distribution in the cavity. The laser light is distributed over a volume of $7 \times 25 \times 176$ mm³, illuminating most of the muon stop volume of $5 \times 12 \times 200$ mm³. The laser frequency is not continuously scanned, but kept fixed over several hours. Initially, during the search for the resonance such a measurement block is typically 5 h long to establish a statistically significant excess of laser-induced events over background. After finding the resonance, we alternate between both sides of it every two hours, interlaced with background measurements. During these measurement blocks, the continuous-wave Ti:Sa laser is locked to a Fabry–Pérot cavity and its frequency is recorded by a High Finesse WS7 wavelength meter, both of them calibrated by saturation spectroscopy in caesium and krypton. Simple absorption spectroscopy is used to measure the average frequency shift (chirp) between the pulses delivered by the Ti:Sa

oscillator and its seeding light of −0.1 ± 0.05 GHz. After the laser excitation of the 2S → 2P transition, we detect the Lyman-α X-ray from the subsequent decay to the ground state via 20 large-area ($14 \times 14$ mm²) avalanche photodiodes[31,57] (LAAPDs) arranged above and below the stop volume, covering a fraction of about 30% of the full solid angle. We also detect delayed electrons from muon decay with the LAAPDs and with plastic scintillators radially arranged around the muon stop volume, outside of the target.

The LAAPDs detect about 50,000 signals (events) per hour, of which there are about 8 events from successful excitation to the 2P state, if the laser is on resonance. We exclude most of the background by selecting only events with an energy in a range of [7.9, 8.5] keV in a roughly 230-ns-wide time window coincident with the laser pulse in the cavity followed by a muon decay electron. We apply a waveform analysis[31] of the digitized LAAPD signals, which allows us to increase the energy resolution by about a factor of two to 16% full-width at half-maximum (FWHM) at 8.2 keV (relative to a standard approach based on the area of the pulse) and distinguish between electrons and X-rays. To improve the electron detection, four plastic scintillators of 5 mm thickness are radially placed around the target to detect the decay electrons curling in the magnetic field. Furthermore, we use the muon entrance detector to discard, with an efficiency >90%, the data when two muons are in the target at the same time.

The number of selected Lyman-α X-rays is normalized to the prompt events to account for fluctuations in the muon beam. This ratio is then plotted versus the laser frequency, as shown in Fig. 2. The obtained resonance is fitted with a line-shape model based on a Lorentz profile, accounting for the saturation level and the measured pulse energy. The fits result in $\chi^2 = 92.3$ for 89 degrees of freedom and $\chi^2 = 17.1$ for 16 degrees of freedom for the 2S → 2P$_{3/2}$ and 2S → 2P$_{1/2}$ transitions, respectively. The significance of this result compared with fitting a flat line is 15$\sigma$ and 11$\sigma$, respectively. We find that a naive fit of a simple Lorentzian without accounting for the pulse energy produces line centres in agreement, within 2 GHz, to the ones from the line-shape model. This implies that the pulse energy asymmetry, left and right of the resonance, is strongly mitigated by our procedure: we adjust the transmission of the $\lambda/2$-polarizer system at the beginning of each measurement block, and scan the resonance by alternating the laser frequency between left and right of the resonance. In fact, we measure a left–right asymmetry of about 1% for both resonances. Such an asymmetry would lead to a shift of 0.5% of the FWHM of 320 GHz, that is, 1.6 GHz (if not accounted for in the line-shape model). This is in agreement with the observed shift of 2 GHz for each of the resonances when fitting a simple Lorentzian. To account for possible systematic effects in the measurement of the laser pulse energy, we quote a conservative total systematic uncertainty of 3 GHz for each of the two resonances.

### Theory

The α-particle charge radius determined from laser spectroscopy of the Lamb shift in muonic helium-4 ions depends crucially on the theory of the 2S−2P energy splitting. Here we briefly explain the origin of the various terms of the equation (1), which summarizes all known theory contributions to the Lamb shift in the muonic helium-4 ion. It differs slightly from our previously published formula, equation 29 in ref. [23], which reads

$$
\begin{aligned}
\Delta E_{2P_{1/2}-2S}^{\text{theo}} = {} & 1{,}668.489(14)\ \text{meV} \\
& - 106.354(8)\ \text{meV fm}^{-2} \times r_\alpha^2 + 0.078(11)\ \text{meV} \\
& + 9.340(250)\ \text{meV}.
\end{aligned}
\tag{10}
$$

Recently, some of the 2PE and 3PE contributions with nuclear structure were calculated in ref. [27], and a large inelastic three-photon contribution was identified for muonic D. Unfortunately, this contribution has not yet been calculated for muonic helium-4. We estimate this term and

take it into account with a 100% uncertainty, as detailed below. The updated theory reads

$$\Delta E_{2P_{1/2}-2S}^{\text{theo}} = 1{,}668.489(14) \text{ meV}$$
$$- 106.220(8) \text{ meV fm}^{-2} \times r_\alpha^2 + 0.0112 \text{ meV}$$
$$+ 9.340(250) \text{ meV} \qquad (11)$$
$$- 0.150(150) \text{ meV}.$$

Here we detail the changes leading to equation (11) (same as equation (1)). Note that the effect on the extracted $r_\alpha$ is small: the central value shifts by 0.6 standard deviations, and the theory uncertainty increases by 15%.

The first term in both equations, 1,668.489(14) meV, is the sum of radiative, relativistic and recoil corrections independent of the nuclear structure[58–62]. Of this term, 99.8% (1,665.773 meV) is given by the one-loop electron vacuum polarization. The one-loop vacuum polarization in muonic atoms is vastly enhanced compared with the self-energy (which dominates in ordinary atoms) given its short-range interaction.

Physically, the large mass of the orbiting muon leads to relatively large momenta of the photons exchanged with the nucleus, facilitating the creation of virtual electron–positron pairs, which are much lighter than the muon. There is no corresponding term in ordinary H, because there are no charged particles that are lighter than an electron. Muonic vacuum polarization (with a muon pair in the loop) exists as well, and it can be calculated with a simple rescaling of the electron vacuum polarization result for ordinary H (the mass of the orbiting lepton is equal to the mass of the virtual particle in the loop). For comparison, the sum of muonic vacuum polarization and muon self-energy amounts to −11.106 meV (ref. [23]).

The second term, proportional to the square of the α-particle charge radius $r_\alpha$, accounts for the energy-level shifts caused by the finite size of the nucleus, including some mixed radiative–finite-size corrections to the one-photon exchange. Note that the proper relativistic definition of $r_\alpha$ is given by the slope of the form factor $r_\alpha^2 = -6G_E'(Q^2 = 0)$ (refs. [55,63]), which in a non-relativistic approximation (that holds well for $(\mu^4\text{He})^+$) corresponds to the r.m.s. radius of the charge distribution $r_\alpha \approx \sqrt{\langle r^2 \rangle}$. In the older equation (10), the finite-size contribution contains also the parts of the $\alpha(Z\alpha)^5$ (radiative correction to the 2PE) and $(Z\alpha)^6$ (3PE, elastic part) contributions ($\alpha$, fine structure constant) that can be parameterized with $r_\alpha^2$ (#r2 and #r3 of ref. [23], respectively).

The next, small, term in equation (10) of 0.078(11) meV is a sum of the parts of the $\alpha(Z\alpha)^5$ and $(Z\alpha)^6$ contributions (3PE, elastic contribution) that cannot be simply parameterized as being proportional to $r_\alpha^2$. These parts (#r2′ and #r3′ in ref. [23]) depend on various moments of the charge distribution ($\langle 1/r \rangle$, $\langle \log r \rangle$, $\langle r^3 \rangle$ and so on)[24,64]. Because these terms are small, they have been calculated for the scattering value of $r_\alpha$ assuming a charge distribution[58,64] or, equivalently, using various parameterizations of the form factor[59].

The fourth term is the 2PE contribution resulting from the sum of the third Zemach moment contribution extracted from elastic electron scattering data on helium[25], and the polarizability contribution computed using a state-of-the-art ab initio few-nucleon approach[36]. Recently, two- and three-photon contributions in muonic H, D, helium-3 and helium-4 have been addressed in ref. [27]. A full calculation of both the elastic and inelastic 3PE exists only for muonic D. Unexpectedly, the inelastic part of the 3PE in muonic D turned out to be of similar size as the elastic part, but of opposite sign. This leads to a substantial cancellation for the full 3PE in muonic D. For muonic H, by contrast, the inelastic part is estimated to be much smaller than the elastic part.

This may be understood considering that the proton is much 'stiffer' than the 'softer' deuteron, that is, the first excited state of the proton (the $\Delta$ resonance), is 300 MeV above the proton ground state, whereas the binding energy of the deuteron is only 2.2 MeV. For comparison, the α-particle binding energy is 28 MeV.

Unfortunately, the inelastic 3PE for muonic He has not yet been calculated[27]. However, it can be estimated starting from the calculated elastic part of the 3PE in $(\mu^4\text{He})^+$, and applying a similar cancellation as observed in muonic D. Starting from the calculated[27] elastic 3PE in $(\mu^4\text{He})^+$ of

$$\Delta(3\text{PE, elastic}) = -0.3048 \text{ meV}, \qquad (12)$$

we assume the inelastic part to be of opposite sign, and assign it a range between 0 and 1.0 times the full elastic part (within 1 standard deviation). Hence, we estimate the inelastic part to be

$$\Delta(3\text{PE, inelastic}) = -0.3048 \times (-0.5) \times (1\pm1)$$
$$= +0.150(150) \text{ meV}, \qquad (13)$$

that is, with a conservative 100% uncertainty. For comparison, in muonic D, the calculated inelastic 3PE is −1.3 times the elastic 3PE. Note that the polarizability of the loosely bound deuteron (2.2-MeV binding energy) is much larger than for the much tighter-bound α particle (28-MeV binding energy). For the even tighter-bound proton (300-MeV energy required to reach the first excited state) the inelastic 3PE is expected to be very small. One thus expects that the ratio of inelastic-to-elastic 3PE in $(\mu^4\text{He})^+$ is between these two extreme cases observed in muonic H and muonic D.

The total 3PE resulting from the sum of elastic and inelastic contribution becomes

$$\Delta(3\text{PE, elastic + inelastic}) = -0.150(150) \text{ meV}. \qquad (14)$$

Such an estimate and its uncertainty has been approved by K. Pachucki (personal communication), one of the authors of ref. [27]. Note that the $1.6\sigma$ band of this sum includes the extreme cases, namely both a vanishing inelastic 3PE and an inelastic 3PE being −1.3 times the elastic part, respectively.

Given the updated value of the elastic 3PE for $(\mu^4\text{He})^+$ of ref. [27], and the estimate of the inelastic part given above, we have updated the theoretical prediction of the Lamb shift in muonic He, as given in equation (11). The first term of 1,668.489(14) meV is the same as in equation (10). The coefficient in front of the $r_\alpha^2$ term in equation (11) differs slightly from the coefficient in equation (10), because the part of the elastic 3PE contribution that can be expressed as being proportional to $r_\alpha^2$ (the #r3 term in ref. [23])

$$\text{#r3} = -0.1340(30) \text{ meV fm}^{-2} \times r_\alpha^2 = -0.377(8) \text{ meV} \qquad (15)$$

has been moved to the last term of equation (11).

Similarly, also the third term of 0.0112 meV in equation (11) differs slightly from the third term in equation (10), because the part of the 3PE (#r3′ in ref. [23])

$$\text{#r3}' = 0.067(11) \text{ meV} \qquad (16)$$

that cannot be simply parameterized using $r_\alpha^2$, has also been included in the last term of equation (11).

The fourth term in equation (11), representing the total 2PE contribution, is identical to the fourth term in equation (10).

The last term in equation (11) represents the total 3PE, where the #r3 and #r3′ included in equation (10) have been updated by the elastic part newly calculated in ref. [27], and where the inelastic part has been estimated using the arguments given above.

Using the updated theoretical prediction given in equation (11) (that is, equal to equation (1)), we find an α-particle charge radius from our measurement of $r_\alpha = 1.67824(13)_{\text{exp}}(82)_{\text{theo}}$ fm. For comparison, using

equation (10), which neglects the inelastic 3PE contribution, yields a radius of $r_\alpha = 1.67779(13)_{exp}(71)_{theo}$ fm, that is, a change of 0.00045 fm, corresponding to 0.6 standard deviations.

## Data availability
Reasonable requests for data should be addressed to R.P.

## Code availability
Reasonable requests for computer code should be addressed to the corresponding authors.

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

**Acknowledgements** This work was performed at HIPA at PSI. We thank the accelerator and beamline support groups for excellent conditions. We also thank L. Simons, U. Röser, M. Nüssli, H. v. Gunten, B. Zehr, W. Lustermann, A. Gendotti, A. Müller, F. Barchetti, B. van den Brandt, P. Schurter, M. Horisberger, A. Weber, S. Spielmann-Jäggi, U. Greuter, P.-R. Kettle, S. Ritt, W. Simons, K. Linner, H. Brückner, K. S. E. Eikema and the PSI, ETH and MPQ workshops and support groups for their help. We acknowledge the support of the following grants: European Research Council (ERC) through StG. 279765 and CoG. 725039, Deutsche Forschungsgemeinschaft (DFG, German Research Foundation) under Germany's Excellence Initiative EXC 1098 PRISMA (194673446) and Excellence Strategy EXC PRISMA+ (390831469), EU Horizon 2020 innovation programme STRONG-2020 (grant agreement number 824093), DFG_GR_3172/9–1, MOST of Taiwan under contract number 106-2112-M-007 -021 -MY3, Fundação para a Ciência e a Tecnologia (FCT), Portugal, and FEDER through COMPETE in the framework of project numbers PTDC/FIS-NUC/0843/2012, PTDC/FIS-NUC/1534/2014, PTDC/FIS-AQM/29611/2017, PEstOE/FIS/UI0303/2011, PTDC/FIS/117606/2010 and UID/04559/2020 (LIBPhys), contract numbers SFRH/BPD/92329/2013, SFRH/BD/52332/2013, SFRH/BD/66731/2009 and SFRH/BPD/76842/2011, and by SNF 200021L_138175, SNF 200020_159755 and SNF 200021_165854, as well as the ETH-FAST initiative as part of the NCCR MUST programme.

**Author contributions** J.J.K., K.S., M.A.A., F.B., T.-L.C., M.D., S.G., T.G., T.W.H., L.J., K.K., Y.-W.L., B.N., F.N., D.T., J.V., A.V., B.W., R.P., A.A. and F.K. designed, built and operated parts of the laser system. J.J.K., K.S., F.D.A., M.D., L.M.P.F., B.F., A.L.G., M.H., K.K., A.K., Y.-W.L., J.M., C.M.B.M., F.M., T.N., F.N., J.M.F.S., J.P.S., D.T., J.F.C.A.V., R.P., A.A. and F.K. planned, built and set up the various detectors of the experiment. J.J.K., K.S., F.D.A., P.A., D.S.C., M.D., L.M.P.F., B.F., J.G., J.H., M.H., K.K., A.K., J.M., F.M., J.P.S., C.I.S., D.T., R.P., A.A. and F.K. designed, built, set up and operated the muon beam line. J.J.K., K.S., F.D.A., M.D., L.M.P.F., B.F., A.K., J.M., C.M.B.M., F.M., F.N., J.M.F.S., D.T., R.P., A.A. and F.K. designed and implemented the electronics used in the experiment. J.J.K., P.A., M.D., A.K., J.P.S., J.V., R.P. and A.A. set up the computing infrastructure, wrote software and realized the data acquisition system. J.J.K., K.S., F.D.A., P.A., D.S.C., A.J.D., M.D., L.M.P.F., B.F., S.G., J.G., A.L.G., J.H., P.I., L.J., A.K., Y.-W.L., J.M., C.M.B.M., B.N., F.N., J.M.F.S., J.P.S., C.I.S., D.T., J.F.C.A.V., R.P., A.A. and F.K. took part in the months-long data taking runs. J.J.K., P.A., M.D., B.F., P.I., J.P.S., D.T., R.P., A.A. and F.K. did work on theory. J.J.K., K.S., M.D., L.M.P.F., P.I., C.M.B.M., F.N., J.M.F.S., R.P., A.A. and F.K. analysed the data J.J.K., R.P., A.A. and F.K. drafted this manuscript. The manuscript was then read, improved and finally acknowledged by the other authors.

**Funding** Open access funding provided by Max Planck Society.

**Competing interests** The authors declare no competing interests.

**Additional information**
**Correspondence and requests for materials** should be addressed to J.J.K., R.P. or A.A.
