## [Peer Review File · Nature]

Supplementary information

Measuring the α -particle charge radius with muonic helium-4 ions

In the format provided by the authors and unedited

Peer Review File

Manuscript Title: Measuring the alpha particle charge radius with muonic ^4He ions

Publishers Note:

Please note that supplementary materials detailed in the below comments were moved to the methods section of the article.

Editorial Notes:**Redactions – reviewer opt-out**

Parts of this Peer Review File have been redacted as indicated as we could not obtain permission to publish the reports of a reviewer.

Reviewer Comments & Author Rebuttals**Reviewer Reports on the Initial Version:**

Referee #1 (Remarks to the Author):

[REDACTED]

Referee #2 (Remarks to the Author):

The Authors (The CREMA collaboration) performed the excellent laser spectroscopy measurement of 2S–2P lines in the muonic helium-4 ion. The accuracy of measured transitions is sufficient to determine the rms charge radius of the alpha particle the most accurately. There is no doubt that the alpha particle is a very important nucleus of wide interest in various fields i.e. few-body nuclear, elementary and atomic physics including tests of the Standard Model of fundamental interactions or examine the nuclear structure with high precision. This work can be considered potentially as an important achievement, exceeding the current limits of laser spectroscopy on muonic systems in determining the basic component of matter and deserving of visibility in the best scientific journals, just like Nature.

Indeed, the final result here is finding the rms charge radius of the alpha particle $r_{\alpha} = 1.67779(72)$ fm and the final conclusion is that the value is in perfect agreement with the value obtained from electron scattering, but it is much more precise. However, in addition to the measurement data, a valid theory must be provided. The summary of various contributions compiled by the Authors from the literature can be found in equation (1) including the crucial term proportional to r_{α}^2 . Here I see the only, but serious problem with the theory used as the basis of this work.

1) As noted by the Authors in lines 96-98 "the potentially important inelastic three-photon contribution in $(\mu^4\text{He})^+$ has not been calculated and is therefore not included in Eq. (1)." We know from the muonic deuterium that the magnitude of elastic and inelastic contributions are

comparable, and that they cancel each other significantly e.g. PRA 97, 062511 (2018) (Table II). The same one can expect for muonic helium ion. Thus, the neglect of the inelastic three-photon contribution must be justified at least by estimating this effect. Moreover, we cannot exclude at this point that the final conclusion will be completely different from the one stated as the main achievement of this work, i.e. consistency with the electron scattering result but the higher accuracy. The accuracy of r_α may now be significantly overstated by 1-2 orders of magnitude. I strongly recommend the Authors to clarify this issue, because the experiment is really great.

2) The energy difference in Eq. 1) should depend not only on r_α^2 but also on $r_\alpha^2 \text{Log}[m r_\alpha]$. It would be good for some comment to be made on how such terms have been treated in Eq. 1).

To summarize my evaluation of this work, I have a problem with the claim that the present state of the project has enough rigorous justification of the accuracy of r_α due to insufficient theory. It is necessary at least to estimate the inelastic three photon contribution, which unfortunately can significantly change the meaning of the project.

Author Rebuttals to Initial Comments:

We thank the Referees for their valuable comments on our submitted manuscript and address the individual points below. The changes we made due to the referees' comments are highlighted in the paper with colored text and the important parts are also shown here in bold letters.

(Q1) Referee #1 (Remarks to the Author):

[REDACTED]

(A1) We agree with the comment of the referee. Hence, we reformulated the end of our abstract in the following way (line 21-26):

This agreement also constraints several Beyond Standard Model theories proposed to explain the proton radius puzzle²⁻⁵, in line with recent determinations of the proton charge radius⁶⁻⁹, and establishes spectroscopy of light muonic atoms and ions as a precise tool for studies of nuclear properties.

(Q2)

[REDACTED]

(A2) We thank the author for this important remark and decided to ask Prof. K. Pachucki from Warsaw University for an estimate of this difficult to calculate contribution. He is one of the leading experts in this field and one of the authors of the paper which calculates the inelastic three-photon contribution for muonic hydrogen and deuterium, where the calculation is not as complicated as in muonic helium-4. Since a calculation of the inelastic part of muonic helium-4 does not exist at this moment, he supports to include a value of 0.15 meV with 100% uncertainty and opposite sign to the

elastic contribution as a conservative estimate. This value accounts for the possible complete cancellation of this term with its elastic counterpart within one standard deviation. We follow his advice and add this term to Eq.(1). This results in an updated value of the charge radius of $r_\alpha = 1.67824(83)$ fm (old $r_\alpha = 1.67779(72)$ fm), which is within 0.6 standard deviations of the previously stated value, now with a 15% larger uncertainty. The conclusions of our paper stay therefore untouched. Eq.(1) and all related values have been updated. Due to a comment of the second Referee we also reorganized the different contributions in Eq.(1) (between line 86 and 87) and added an explanatory Supplement.

(Q3)

[REDACTED]

(A3) In ordinary He⁺ ions and other hydrogen-like atoms the dominating QED effect is the self energy contribution. The one-loop electron vacuum polarization is a much smaller effect. In muonic atoms, however, the situation is different: On one hand, there is the muon vacuum polarization, which is of same relative size as the electron vacuum polarization in ordinary hydrogen-like atoms. On the other hand there is the electron vacuum polarization in a muonic atom, namely a vacuum polarization term in which the created electron-positron pair is of much smaller mass than the bound muon. Such a term doesn't exist in ordinary atoms, since there is no lepton lighter than the electron. The electron vacuum polarization in muonic atoms is the dominating QED effect and in this case contributing as much as 99.8% to the nuclear-structure-independent Lamb shift (first term in Eq. (1)).

This is now mentioned briefly in the main text:

The Lamb shift is dominated by pure QED effects, in particular vacuum polarization, which is vastly enhanced in muonic atoms²³, see Supplementary Material (SM), [...]

and more detailed in the Supplement, see Footnote a therein

(Q4)

[REDACTED]

(A4) We agree that the previous formulation was not correct and we have reformulated text accordingly (lines 48-61):

[...] because the required combination of sufficiently precise measurements and theory calculations exists only for atomic H and D (atomic number $Z = 1$). For elements with $Z > 1$, laser spectroscopy has yielded only differences of charge radii within an isotopic chain¹³⁻¹⁸ by measuring the same atomic transition in various isotopes to eliminate the common energy shifts related with the interaction among electrons. Indeed for the determination of absolute radii from He atoms (three body system with two electrons), theory is not yet advanced enough¹⁹. Sufficiently precise

experiments with the H-like He^+ ion, where the two-body theory of hydrogen is applicable, will be available in the future^{20,21}.

(Q5) Referee #2 (Remarks to the Author):

The Authors (The CREMA collaboration) performed the excellent laser spectroscopy measurement of 2S–2P lines in the muonic helium-4 ion. The accuracy of measured transitions is sufficient to determine the rms charge radius of the alpha particle the most accurately. There is no doubt that the alpha particle is a very important nucleus of wide interest in various fields i.e. few-body nuclear, elementary and atomic physics including tests of the Standard Model of fundamental interactions or examine the nuclear structure with high precision. This work can be considered potentially as an important achievement, exceeding the current limits of laser spectroscopy on muonic systems in determining the basic component of matter and deserving of visibility in the best scientific journals, just like Nature.

Indeed, the final result here is finding the rms charge radius of the alpha particle $r_{\alpha} = 1.67779(72)$ fm and the final conclusion is that the value is in perfect agreement with the value obtained from electron scattering, but it is much more precise. However, in addition to the measurement data, a valid theory must be provided. The summary of various contributions compiled by the Authors from the literature can be found in equation (1) including the crucial term proportional to r_{α}^2 . Here I see the only, but serious problem with the theory used as the basis of this work.

1) As noted by the Authors in lines 96-98 "the potentially important inelastic three-photon contribution in $(\mu^4\text{He})^+$ has not been calculated and is therefore not included in Eq. (1)." We know from the muonic deuterium that the magnitude of elastic and inelastic contributions are comparable, and that they cancel each other significantly e.g. PRA 97, 062511 (2018) (Table II). The same one can expect for muonic helium ion. Thus, the neglect of the inelastic three-photon contribution must be justified at least by estimating this effect. Moreover, we cannot exclude at this point that the final conclusion will be completely different from the one stated as the main achievement of this work, i.e. consistency with the electron scattering result but the higher accuracy. The accuracy of r_{α} may now be significantly overstated by 1-2 orders of magnitude. I strongly recommend the Authors to clarify this issue, because the experiment is really great.

(A5) Indeed. This is our main change, see answer A2 above.

(Q6)

2) The energy difference in Eq. 1) should depend not only on r_{α}^2 but also on $r_{\alpha}^2 \text{Log}[m r_{\alpha}]$. It would be good for some comment to be made on how such terms have been treated in Eq. 1).

(A6) These terms have been taken into account in the theory, but were not mentioned explicitly. Following the advice we have reorganized the terms in Eq.(1), included the corresponding explanations below the equation (lines 87-109)

The second term is the finite-size effect. It is proportional to the square of the alpha-particle rms charge radius and includes mixed radiative - finite-size contributions. The next, small term is implicitly radius-dependent but can not be parametrized as being proportional to r^2 . Because this term is small it is sufficient to calculate it using electron scattering results³¹.

and added more detail in the Supplement.

(Q7)

To summarize my evaluation of this work, I have a problem with the claim that the present state of the project has enough rigorous justification of the accuracy of r_{α} due to insufficient theory. It is necessary at least to estimate the inelastic three photon contribution, which unfortunately can significantly change the meaning of the project.

(A7) We believe that this concern is now allayed by including a conservative estimate of the inelastic 3PE. The uncertainty of the charge radius increases by only 15%.

Reviewer Reports on the First Revision:

Referee #1 (Remarks to the Author):

[REDACTED]

Referee #2 (Remarks to the Author):

This is my second approach to reviewing the manuscript "The alpha particle charge radius from laser spectroscopy of the muonic helium-4 ion" by Julian J. Krauth et al. The authors have carefully followed the referees' recommendations. In particular, the Authors conservatively estimated the previously neglected effect of the inelastic three-photon contribution in $(\mu^4\text{He})^+$. This turned out to be important, but did not alter the main result (r_{α}) from the previous version of the manuscript in a way that would imply any significant change in the final conclusions. It is undoubtedly a very comfortable situation. I do not have any further comments to the resubmitted version.

I uphold my evaluation of this work, it represents a research of very high quality. I recommend publishing the manuscript in Nature.

Referee #3 (Remarks to the Author):

Dear Editor,

it was a pleasure to read the manuscript "The alpha particle charge radius from laser spectroscopy of the muonic helium-4 ion" by Krauth et al.. I found the paper written in a very clear and comprehensible way and I believe it will be understandable at least for the complete physics community and to some extent also to a wider range of interested people from neighboring fields. The paper reports on the successful accomplishment of a very challenging experiment, namely laser spectroscopy on muonic helium (4He) to determine the nuclear charge radius of the alpha particle. Such an experiment has already been tried more than 40 years ago, but the claim of the experimenters to have spotted the transition has been highly debated in the community and many doubts were raised and it is now finally disproved by the results presented here.

The new experiment has doubtlessly observed both fine-structure transitions and - by combining it with state-of-the-art atomic theory and QED - was able to extract the nuclear charge radius of this very important nucleus with unprecedented accuracy.

The result seems to be less exciting than the proton-radius result that produced the "proton radius puzzle" but it has far-reaching consequences since it can be used in multiple ways in a number of fields reaching from fundamental strong- interaction physics in lattice QCD, across nuclear structure physics of stable and short-lived isotopes, to QED tests and constraints for beyond-standard-model physics. This is very well explained and discussed in the manuscript. Besides confirming the elastic electron scattering result - contrary to the hydrogen and deuterium case reported earlier - it will provide the opportunity to determine the Rydberg constant more precisely once the $1s-2s$ transition in the hydrogen-like He^+ system has been measured for which experiments are currently being prepared. Moreover, many speculations about the role of beyond-standard-model physics in muonic atoms that arose after the observation in hydrogen can now be constrained in size if not excluded.

For these reasons I consider this as an important progress in our field and believe the manuscript is worth being published in Nature. I find it actually an example how to present and discuss physics results. The supplementary material is supportive in understanding the theory. The results in the main paper is presented in a way that one can easily reproduce all calculations starting from the experimentally observed transition frequencies down to the charge radius including all uncertainties.